# Motion Policy Networks

**Adam Fishman** [1,*]  **Adithyavairavan Murali** [2]  **Clemens Eppner** [2]
**Bryan Peele** [2]   **Byron Boots** [1,2]   **Dieter Fox** [1,2]
[1]University of Washington   [2]NVIDIA
{afishman, bboots, fox}@cs.washington.edu
{admurali, ceppner, bpeele}@nvidia.com

https://mpinets.github.io

**Abstract:** Collision-free motion generation in unknown environments is a core
building block for robot manipulation. Generating such motions is challenging
due to multiple objectives; not only should the solutions be optimal, the motion
generator itself must be fast enough for real-time performance and reliable enough
for practical deployment. A wide variety of methods have been proposed ranging
from local controllers to global planners, often being combined to offset their
shortcomings. We present an end-to-end neural model called Motion Policy Net-
works (MπNets) to generate collision-free, smooth motion from just a single depth
camera observation. MπNets are trained on over 3 million motion planning prob-
lems in more than 500,000 environments. Our experiments show that MπNets are
significantly faster than global planners while exhibiting the reactivity needed to
deal with dynamic scenes. They are 46% better than prior neural planners and
more robust than local control policies. Despite being only trained in simulation,
MπNets transfer well to the real robot with noisy partial point clouds. Videos and
code are available at https://mpinets.github.io.

**Keywords:** Motion Control, Imitation Learning, End-to-end Learning

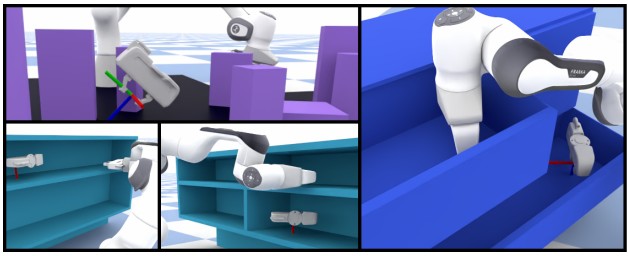 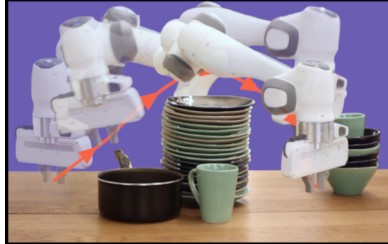

Figure 1: MπNets are trained on a large dataset of synthetic demonstrations (*left*) and can solve
complex motion planning problems using raw point cloud observations (*right*).

## 1 Introduction

Generating fast and legible motions for a robotic manipulator in unknown environments is still an
open problem. Decades of research have established many well-studied algorithms, but there are
two practical issues that prevent motion planning methods from being widely adopted in industrial
applications and home environments that require real-time control. First, it is challenging for any
single approach to satisfy multiple planning considerations: speed, completeness, optimality, ease-
of-use, legibility (from the perspective of a human operator), determinism, and smoothness. Second,
existing approaches enforce strong assumptions about the visual obstacle representations—such as
accurate collision checking in configuration space [1] or the availability of a gradient [2, 3, 4]—and
hence require expensive intermediate processing of sensor readings to operate in novel scenes.

---

*Work done while the author was an intern at NVIDIA.

6th Conference on Robot Learning (CoRL 2022), Auckland, New Zealand.

Some planners, such as RRT [5], are useful to quickly find a feasible path. Other methods are useful to find an optimal path, either deterministically on discrete graphs [6, 7] or asymptotically in continuous spaces [8, 9]. Some planning techniques use Neural Networks [10, 11] to improve sampling efficiency, thus speeding up the planning process. Optimization-based methods [2, 3] are useful to encode other objectives, such as smoothness, at the expense of guarantees. Going further, recent motion generation frameworks [4, 12] eschew long-term reasoning in exchange for quick local decisions under the assumption that they will lead to globally acceptable paths.

Each of these methods requires a known environment model and perfect state estimation. In practice, one would have to create a scene representation, which could be a static or dynamic mesh, an occupancy grid [11, 13], a signed distance field, etc. Reconstruction systems such as SLAM and KinectFusion [14] have a large system start-up time, require a moving camera to aggregate many viewpoints, and ultimately require costly updates in the presence of dynamic objects. Recent implicit deep learning methods like DeepSDF [15] and NeRF [16] are slow or do not yet generalize to novel scenes. Methods such as SceneCollisionNet [17] provide fast collision checks but have not yet been shown to generalize to challenging environments beyond a tabletop. Other RL-based methods learn a latent representation from observations but have only been applied to simple 2D [18, 19] or 3D [20] environments in simulation.

We present *Motion Policy Networks (MπNets)*, a novel method for learning an end-to-end policy for motion planning. Our approach circumvents the challenges of traditional motion planning and is flexible enough to be applied in unknown environments. Our contributions are as follows:

- We present a large-scale effort in neural motion planning for manipulation. Specifically, we learn from over 3 million motion planning problems across over 500,000 instances of three types of environments, nearly 300x larger than prior work [10].

- We train a reactive, end-to-end neural policy that operates on point clouds of the environment and moves to task space targets while avoiding obstacles. Our policy is significantly faster than other baseline configuration space planners and succeeds more than local task space controllers.

- On our challenging dataset benchmarks, we show that MπNets is nearly 46% more successful at finding collision-free paths than prior work [10] without even needing the full scene collision model.

- Finally, we demonstrate *sim2real* transfer to real robot partial point cloud observations.

## 2 Related Work

**Global Planning:** Robotic motion planning typically splits into three camps: search, sampling, and optimization-based planning. Each algorithmic family has benefits and drawbacks. Search-based planning algorithms, such as A* [6, 7, 21] are complete, fast, and optimal in discrete domains. Sampling-based methods, e.g. [5, 8], operate in continuous domains, but are only probabilistically complete, *i.e.* find a solution with probability 1. Some such methods are also asymptotically optimal [8, 22, 9], but within practical time limitations produce sub-optimal—and sometimes erratic—paths. Motion Optimization [2, 3, 23] can produce smooth paths to a goal but is prone to local minima. Without careful system design, often on a per-task basis, Motion Optimization can fail to find the optimal solution or sometimes any solution at all. See Appendix A for a deeper discussion on the varying advantages and disadvantages of each algorithmic family.

**Local Control:** In contrast to global planners, local controllers have long been applied to create collision-free motions [24, 25, 4, 12]. While they prioritize speed and smoothness, they are highly local and may fail to find a valid path in complex environments. We demonstrate in our experiments that MπNets are more effective at producing convergent motions in these types of environments, including in dynamic and in partially observed settings.

**Imitation Learning:** Imitation Learning [26] can train a policy from expert demonstrations with limited knowledge of the expert's internal model. For motion planning problems, we can apply imitation learning and leverage a traditional planner as the expert demonstrator—with a perfect model of the scene during training—and learn a policy that forgoes the need for an explicit scene model at test time. Popular imitation learning methods include Inverse Reinforcement Learning [27, 28, 29] and Behavior Cloning [30, 31]. See Appendix A for a more detailed discussion of these methods and their trade-offs. Our method seeks to overcome the common challenges of Behavior

Cloning by specifically designing a learnable expert, increasing the scale and variation of the data, and using a sufficiently expressive policy model.

**Neural Motion Planning:** Many deep planning methods [11, 32, 33, 34] seek to learn efficient samplers to speed up traditional planners. Motion Planning Networks (MPNets) [10] learn to directly plan through imitation of a standard sampling-based RRT* planner [8] and is used in conjunction with a traditional planner for stronger guarantees. While these works greatly improve the speed of the planning search, they have the same requirements as a standard planning system: targets in configuration space and an explicit collision checker to connect the path. Our work operates based on task-space targets and perceptual observations from a depth sensor without explicit state estimation.

Novel architectures have been proposed, such as differentiable planning modules in Value Iteration Networks [19], transformers by Chaplot et al. [35] and goal-conditioned RL policies [36]. These methods are challenging to generalize to unknown environments or have only been shown in simple 2D [18] or 3D settings [20]. In contrast, we illustrate our approach in the challenging domain of controlling a 7 degrees of freedom (DOF) manipulator in unknown, dynamic environments.

# 3 Learning from Motion Planning

## 3.1 Problem Formulation

M$\pi$Nets expect two inputs, a robot configuration $q_t$ and a segmented, calibrated point cloud $z_t$. Before passing $q_t$ through the network, we normalize each element to be within $[-1, 1]$ according to the limits for the corresponding joint. We call this $q_t^{\|\cdot\|}$. The point cloud is always assumed to be calibrated in the robot's base frame, and it encodes three segmentation classes: the robot's current geometry, the scene geometry, and the target pose. Targets are inserted into the point cloud via points sampled from the mesh of a floating end effector placed at the target pose.

The network produces a displacement within normalized configuration space $\dot{q}_t^{\|\cdot\|}$. To get the next predicted state $\hat{q}_{t+1}$, we take $q_t^{\|\cdot\|} + \dot{q}_t^{\|\cdot\|}$, clamp between $[-1, 1]$, and unnormalize. During training, we use $\hat{q}_{t+1}$ to compute the loss, and when executing, we use $\hat{q}_{t+1}$ as the next position target for the robot's low-level controller.

## 3.2 Model Architecture

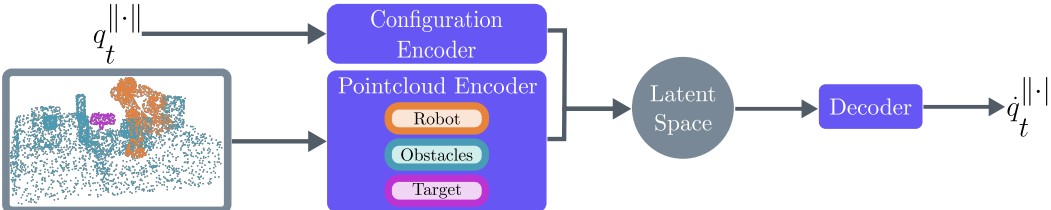

Figure 2: M$\pi$Nets encodes state as a normalized robot configuration and segmented point cloud with three classes for the robot, the obstacles, and the target. The policy outputs a displacement in normalized joint space, which can then be applied to the input before unnormalizing to get $q_{t+1}$.

The network consists of two separate encoders, one for the point cloud and one for the robot's current configuration, as well as a decoder, totaling 19M parameters. Our neural policy architecture is visualized in Fig. 2. We use PointNet++ [37] for our point cloud encoder. PointNet++ learns a hierarchical point cloud representation and can encode a point cloud's 3D geometry, even with high variation in sampling density. PointNet++ architectures have been shown to be effective for a variety of point cloud processing tasks, such as segmentation [37], collision checking [17], and robotic grasping [38, 39]. Additionally, PointNet++ includes PointNet as a subcomponent. PointNet is effective at processing partially observed point clouds, even when trained exclusively with fully-observed scenes [40]. The robot configuration encoder and the displacement decoder are both fully connected multilayer perceptrons. We discuss the architecture in detail in Appendix D.

## 3.3 Loss Function

The network is trained with a compound loss function with two constituent parts: a behavior cloning loss to enforce accurate predictions and a collision loss to safeguard against catastrophic behavior.

**Geometric Loss for Behavior Cloning**  To encourage alignment between the prediction and the expert, we compute a geometric loss across a set of $1,024$ fixed points along the surface of the robot.

$$L_{\mathrm{BC}}(\hat{\Delta}q_t) = \sum_i \|\hat{x}_{t+1}^i - x_{t+1}^i\|_2 + \|\hat{x}_{t+1}^i - x_{t+1}^i\|_1, \text{ where } \begin{matrix} \hat{x}_{t+1}^i = \phi^i(q_t + \hat{\Delta}q_t) \\ x_{t+1}^i = \phi^i(q_{t+1}) \end{matrix} \tag{1}$$

$\phi^i(\cdot)$ represents a forward kinematics mapping from the joint angles of the robot to point $i$ defined on the robot's surface. The loss is computed as the sum of the $L1$ and $L2$ distances between corresponding points on the expert and the prediction after applying the predicted displacement. By using both $L1$ and $L2$, we are able to penalize both large and small deviations.

We use a geometric, task-space loss because our goal is to ensure task-space consistency of our policy. Configuration space loss appears in prior work [10], but does not capture the accumulated error of the kinematic chain as effectively (see Appendix K).

**Collision Loss**  In order to avoid collisions–a catastrophic failure–we apply an additional hinge-loss inspired by motion optimization [41].

$$L_{\mathrm{collision}} = \sum_i \sum_j \|h_j(\hat{x}_{t+1}^i)\|_2, \text{ where } h_j(\hat{x}_{t+1}^i) = \begin{cases} -D_j(\hat{x}_{t+1}^i), & \text{if } D_j(\hat{x}_{t+1}^i) \leq 0 \\ 0, & \text{if } D_j(\hat{x}_{t+1}^i) > 0 \end{cases} \tag{2}$$

The synthetic environments are fully-observable during training, giving us access to the signed-distance functions (SDF), $\{D_j(\cdot)\}_j$, of the obstacles in each scene. For a given closed surface, its SDF maps a point in Euclidean space to the minimum distance from the point to the surface. If the point is inside the surface, the function returns a negative value.

## 4 Procedural Data Generation

### 4.1 Large-scale Motion Planning Problems

Each planning problem is defined by three components: the scene geometry, the start configuration, and the goal pose. Our dataset consists of randomly generated problems across all three components, totaling 3.27 million problems in over $575,000$ environments. We have three classes of problems of increasing difficulty: a cluttered tabletop with randomly placed objects, cubbies, and dressers. Representative examples of these environments are shown in Fig. 1. Once we build these environments, we generate a set of potential end-effector targets and corresponding inverse kinematics solutions. We then randomly choose pairs of these configurations and verify if a plan exists between them using our expert pipeline, as detailed further in Sec. 4.2 and in Appendix E.

### 4.2 Expert Pipeline

Our expert pipeline is designed to produce high-quality demonstrations we want to mimic, *i.e.* trajectories with smooth, consistent motion and short path lengths. Here, *consistency* is meant to describe quality and repeatability of an expert planner—see Appendix C for further discussion. We considered two candidates for the expert - the *Global Planner* which is a typical state-of-the-art configuration space planning pipeline [42] and a *Hybrid Planner* that we engineered specifically to generate consistent motion in task space. For both planners, we reject any trajectories that produce collisions, exceed the joint limits, exhibit erratic behavior (*i.e.* high jerk), or that have divergent motion (*i.e.* final task space pose is more than 5cm from the target).

**Global Planner** consists of off-the-shelf components of a standard motion planning pipeline–inverse kinematics (IK) [43], configuration-space AIT* [42], and spline-based, collision-aware trajectory smoothing [44]. For a solvable problem, as the planning time approaches infinity, IK will find a valid solution and AIT* will produce an optimal collision-free path, both with probability 1. Likewise,

with continuous collision checking, the smoother will produce a smooth, collision-free path. In practice, our dataset size goal—we generated 6.54M trajectories across over 773K environments—dictated our computation budget and we tuned the algorithms according to this limit. We attempted IK at most 1,000 times, utilized an AIT* time out of 20s, and employed discrete collision checking when smoothing. Most commonly, the pipeline failed when AIT* timed out or when, close to obstacles, the smoother's discrete checker missed a collision, thereby creating invalid trajectories.

**Hybrid Planner** is designed to produce consistent motion in task space. The planner consists of task-space AIT* [42] and Geometric Fabrics [4]. AIT* produces an efficient end effector path and Geometric Fabrics produce geometrically consistent motion. The end effector path acts as a dense sequence of waypoints for a sequence of Geometric Fabrics, but as the robot moves through the waypoints, the speed can vary. To promote smooth configuration space velocity over the final trajectory, we fit a spline to the path and retime it to have steady velocity. As we discuss in Sec. 5.1, Geometric Fabrics often fail to converge to a target, so we redefine the planning problem to have the same target as the final position of the trajectory produced by the expert. Inspired by [45], we call this technique *Hindsight Goal Revision (HGR)* and demonstrate its importance in Sec. 5.4. Using the *Hybrid Planner*, we generated 3.27 million trajectories across 576,532 environments.

# 5 Experimental Evaluation

We evaluate our method with problems generated from the same distribution as the training set. See Appendix E for more detail on the procedural generation and random distribution. Within the test set, each problem has a unique, randomly generated environment, as well as a unique target and starting configuration. None of the test environments, starting configurations, nor goals were seen by the network during training. Our evaluations were performed on three test sets: a set of problems solvable by the *Global Planner*, problems solvable by the *Hybrid Planner*, and problems solvable by both. Each test set has 1,800 problems, with 600 in each of the three types of environments.

**Quantitative Metrics:** To understand the performance of a policy, we roll it out until it matches one of two termination conditions: 1) the Euclidean distance to the target is within 1cm or 2) the trajectory has been executed for 20 s (based on consultations with the authors of [4] and [12]). We consider the following metrics (see Appendix H for details):

- *Success Rate* - A trajectory is considered a success if its final position and orientation target errors are below 1 cm and 15° respectively and there are no physical violations.

- *Time* - We measure the wall time for each *successful* trajectory. We also measure *Cold Start (CS) Time*, the average time to react to a new planning problem.

- *Rollout Target Error* - The L2 position and orientation error (taken from [46]) between the target and final end-effector pose in the trajectory.

- *Collision Rate* - The rate of fatal collisions, both self and scene collisions

- *Smoothness* - We use Spectral Arc Length (SPARC) [47] and consider a path to be smooth if its SPARC values in joint and end-effector space are below $-1.6$.

## 5.1 Comparison to Methods With Complete State

Most methods to generate motion in the literature assume access to complete state information in order to perform collision checks. In each of the following experiments, we provide each baseline method with an oracle collision checker. When running MπNets, we use a point cloud sampled uniformly from the surface of the entire scene. Results are shown in Table 1.

**Global Configuration Space Planner** The *Global Planner* is unmatched in its ability to reach a target, but this comes at the cost of average computation time (16.46s) compared to MπNets (0.33s). With a global planner, there is no option to partially solve a problem, meaning the Cold Start Time is equal to the planning time. In a real system, optimizers [2, 3, 48] could be used to quickly replan once an initial plan has been discovered. As discussed in Sec. 4.2, the *Global Planner* is theoretically complete but fails in practice on some of the *Hybrid Planner*-solvable problems due to system timeouts and discrete collision checking during smoothing.

| | Soln. Time (s) | CS Time (s) | Success Rate (%) | | | Smooth (%) |
| | | | Global | Hybrid | Both | |
|---|---|---|---|---|---|---|
| Global Planner [42] | $16.46 \pm 0.90$ | $16.46 \pm 0.90$ | 100 | 78.44 | 100 | 51.00 |
| Hybrid Planner | $7.37 \pm 2.23$ | $7.37 \pm 2.23$ | 50.22 | 100 | 100 | 99.26 |
| G. Fabrics [4] | $0.15 \pm 0.09$ | $2.4e\text{-}4 \pm 3e\text{-}5$ | 38.44 | 59.33 | 60.06 | 85.39 |
| STORM [12] | $4.03 \pm 1.89$ | $13.4e\text{-}3 \pm 2.2e\text{-}3$ | 50.22 | 74.50 | 76.00 | 62.26 |
| MPNets [10] | | | | | | |
| *Hybrid Expert* | $4.95 \pm 23.51$ | $4.95 \pm 23.51$ | 41.33 | 65.28 | 67.67 | 99.97 |
| *Random* | $0.31 \pm 3.55$ | $0.31 \pm 3.55$ | 32.89 | 55.33 | 58.17 | 99.96 |
| M$\pi$Nets (Ours) | | | | | | |
| *Global Expert* | $0.33 \pm 0.08$ | $6.8e\text{-}3 \pm 7e\text{-}5$ | 75.06 | 80.39 | 82.78 | 89.67 |
| *Hybrid Expert* | $0.33 \pm 0.08$ | $6.8e\text{-}3 \pm 7e\text{-}5$ | 75.78 | 95.33 | 95.06 | 93.81 |

Table 1: Algorithm performance on problems sets solvable by planner types. All prior methods use state-information and a oracle collision checker while M$\pi$Nets only needs a point cloud

| | | Evaluation Set | |
| | Training Set | MPNets-Style | Hybrid Expert Solvable (Ours) |
|---|---|---|---|
| MPNets [10] | MPNets-Style | 78.70 | 49.89 |
| M$\pi$Nets (Ours) | MPNets-Style | 33.70 | 5.50 |
| MPNets [10] | Hybrid Expert | 88.90 | 65.28 |
| M$\pi$Nets (Ours) | Hybrid Expert | 89.50 | 95.33 |

Table 2: Success rates (%) of our method compared to Motion Planning Networks (MPNets) [10] trained and evaluated on different datasets

**Hybrid End-Effector Space Planner**   Our *Hybrid Planner* struggles with a large proportion of problems solvable by the *Global Planner*. Yet, its solutions are both faster and smoother than the *Global Planner*. Surprisingly, M$\pi$Nets trained with data from the expert *outperformed* the expert on the *Global Planner*-solvable test set. We attribute this to two features: 1) we use strict rejection sampling to reduce erratic and divergent behavior in our expert dataset and train only on the filtered data and 2) our use of Hindsight Goal Revision to turn an imperfect expert into a perfect one.

**Neural Motion Planning**   Motion Planning Networks (MPNets) [10] proposed a similar method for neural motion planning, but there are a few key differences in both problem setup and system architecture. MPNets requires a ground-truth collision checker to connect sparse waypoints, plans in configuration space, and is not reactive to changing conditions. In the architecture, MPNets uses a trained neural sampler within a hierarchical bidirectional planner. The neural sampler is a fully-connected network that accepts the start, goal, and a flattened representation of the obstacle points as inputs and outputs a sample. MPNets guarantees completeness by using a traditional planner as a fallback if the neural sampler fails to produce a valid plan.

In addition to our data, we generated a set of tabletop problems, which we call *MPNets-Style*, akin to the Baxter experiments in [10], in order to fairly compare the two methods. The results of this experiment can be seen in Table 2. M$\pi$Nets requires a large dataset that covers the space of test problems to achieve compelling performance, while MPNets' utilization of a traditional planning system is much more effective with a small dataset or out-of-distribution problems. However, the MPNets architecture does not scale to more complex scenes, even with more data, as we show in Fig. 3. When trained and evaluated on the Hybrid Planner-solvable dataset, MPNets succeeds in 65.28% of the test set, whereas M$\pi$Nets succeeds in 95.33%, thus decreasing the failure rate by 7X. Furthermore, as we show in Table 1, using the MPNets neural sampler trained with the *Hybrid Planner* performs similarly to a uniform random sampler when both are embedded within the bidirectional MPNets planner.

|  | % Env. Coll. | % Self Coll. | % Jnt Viol. | % Within | | | |
|---|---|---|---|---|---|---|---|
|  |  |  |  | 1cm | 5cm | 15° | 30° |
| G. Fabrics [4] | 8.61 | 0.11 | 0.44 | 69.89 | 75.17 | 83.44 | 85.11 |
| STORM [12] | 0.93 | 0.11 | 0.25 | 79.81 | 83.54 | 81.57 | 85.41 |
| MπNets (Ours) |  |  |  |  |  |  |  |
| *Hybrid Expert* | 0.94 | 0.00 | 0.00 | 98.94 | 99.72 | 98.22 | 99.00 |
| *Global Expert* | 13.78 | 0.06 | 0.00 | 98.67 | 99.89 | 97.56 | 99.11 |

Table 3: Failure Modes on problems solvable by both the global and hybrid planners

**Local Task Space Controllers**   Unlike planners, which succeed or fail in a binary fashion, local policies will produce individual actions that, when rolled out, may fail for various reasons. We break down the various failure modes across the set of problems solvable by both experts in Table 3.

STORM [12] and Geometric Fabrics [4] make local decisions that can lead them to diverge from the target in complex scenarios, such as cluttered environments or those with pockets. While STORM, Geometric Fabrics, and MπNets are all local policies, STORM and Geometric Fabrics rely on human tuning to achieve strong performance. Prior environment knowledge alongside expert tuning can lead to phenomenal results, but these parameter values do not generalize. We used a single set of parameters across all test environments just as we used a single set of weights for MπNets. MπNets encodes long-term planning information across a wide variety of environments, which makes it less prone to local minima, especially in unseen environments.

On problems solvable by the *Hybrid Planner*, MπNets ties or outperforms these other methods across nearly all metrics (see Appendix Table 1). On the set of problems solvable by the *Global Planner*, MπNets target convergence rate is consistently higher, while its collision rate (11%) is worse than either STORM (1.94%) or Geometric Fabrics (7.83%) (see Appendix Table 2). Deteriorating performance on out-of-distribution problems is a typical downside of a supervised learning approach such as MπNets. However, this could be improved with a more robust expert, *e.g.* one with the consistency of our *Hybrid Planner* but the success rate of the *Global Planner*, with finetuning, or with DAgger [49].

## 5.2   Importance of the Expert Pipeline

We observed that the choice of the expert pipeline affects the performance of MπNets. We trained three policies: MπNets-G with 6.54M demonstrations from the *Global Planner*, MπNets-H with 3.27M demonstrations from the *Hybrid Planner*, and MπNets-C with 3.27M demonstrations from each. MπNets-C did not exhibit improved performance over either MπNets-H or MπNets-G (see Appendix K for discussion). When evaluated on a test set of problems solvable by the *Global Planner*, MπNets-G shows far better target convergence (97.94% vs. 87.72%) compared to MπNets-H but worse obstacle avoidance (21.94% collision rate vs. 11%). Nonetheless, MπNets-H is significantly better across all metrics when evaluated on problems solved by both experts as shown in Table 3. We hypothesize that an expert combining the properties of these two–the consistency of the *Hybrid Planner* and the generality of the *Global Planner*, would further improve MπNets's performance. We refer to MπNets-H as MπNets throughout the rest of the paper.

## 5.3   Comparison to Methods With Partial Observations

In addition to demonstrating MπNets' performance on a real robot system, we also compared MπNets to the *Global Planner* (AIT* [42]) in a single-view depth camera setting in simulation. We evaluated on the test set of problems solvable by both the *Global* and *Hybrid Planners*. MπNets only has a minor drop in success rate when using a partial point cloud vs. a full point cloud– from 95.06% to 93.22% though the collision rate increases from 0.94% to 3.06% due to occlusions. For this experiment, we compared to the AIT* component of our *Global Planner* alone to minimize false-positive solutions caused by the smoother's discrete collision checker (see discussion in Section 4.2). We used a voxel-based reconstruction akin to the standard perception pipeline packaged with MoveIt [50]. In our implementation, a voxel is filled only if a 3D point is registered within it. On the same test set using the voxel representation, AIT* produces plans with collisions on 16.41%

of problems. In this setting, M$\pi$Nets's collision rate is over $5X$ smaller than that of the *Global Planner*.

## 5.4 Ablations

We perform several ablations to justify our design decisions. All ablations were trained using the *Hybrid Planner* dataset and evaluated on the *Hybrid Planner*-solvable test set. More ablations and details can be found in Appendix K.

**M$\pi$Nets Performance Scales with More Data** As shown in Fig. 3, the performance of M$\pi$Nets continues to improve with more data, although it saturates at 1.1M. Meanwhile, MPNets [10] has constant performance, demonstrating that our architecture is better able to scale with the data.

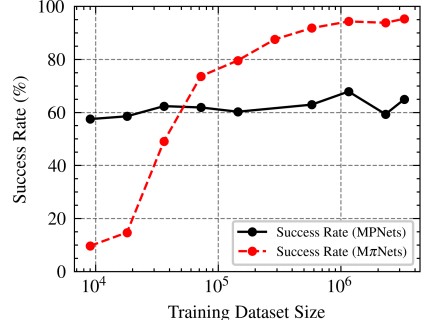

Figure 3: M$\pi$Nets performance continues to increase with more training data, while MPNets performance stays relatively constant

**Robot Point Representation Improves Performance** Instead of representing the robot by its configuration vector, we insert the robot point cloud at the specific configuration. Without this representation, the success rate decreases from $95.33\%$ to $65.06\%$.

**Hindsight Goal Revision Improves Convergence** When trained without *HGR*, *i.e.* with the planner's original target given to the network, we see $58.11\%$ success rate vs. $95.33\%$ when trained with *HGR*. In particular, only $60.28\%$ of trajectories get within 1cm of the target during evaluation.

**Noise Injection Improves Robustness** When we train M$\pi$Nets without injecting noise into the input $q_t$, the policy performance decreases by $10.72\%$.

## 5.5 Real Robot Evaluation

We deployed M$\pi$Nets on a 7-DOF Franka Emika Panda robot with an extrinsically calibrated Intel Realsense L515 RGB-D camera mounted next to it. Depth measurements belonging to the robot are removed and re-inserted using a 3D model of the robot before inference with M$\pi$Nets. We created qualitative open-loop demonstrations in static environments and closed-loop demonstrations in dynamic ones. Rollouts are between 2 and 80 time steps long depending on the control loop frequency. See Appendix M for system details. Results can be viewed at https://mpinets.github.io and the attached video. As can be seen, M$\pi$Nets can achieve *sim2real* transfer on noisy real-world point clouds in unknown and changing scenes.

## 6 Limitations

While M$\pi$Nets can handle a large class of problems, they are ultimately limited by the quality of the expert supervisor and its need for a large, diverse dataset of training examples. Both generating the data and training M$\pi$Nets is computationally intensive, requiring access to equipment that is both economically and environmentally expensive. It will also struggle to generalize to out-of-distribution settings typical of any supervised learning approach. See Appendix N for a deeper discussion on limitations and our plans for future work.

## 7 Conclusion

M$\pi$Nets is a class of end-to-end neural policy policies that learn to navigate to pose targets in task space while avoiding obstacles. M$\pi$Nets show robust, reactive performance on a real robot system using data from a single, static depth camera. We train M$\pi$Nets with what is, as far as we are aware, the largest existing dataset of end-to-end motion for a robotic manipulator. Our experiments show that when applied to appropriate problems, M$\pi$Nets are significantly faster than a global motion planner and more capable than prior neural planners and manually designed local control policies. Code and data are publicly available at https://mpinets.github.io.

**Acknowledgments**

We would like to thank the many people who have assisted in this research. In particular, we would like to thank Mike Skolones for supporting this research at NVIDIA, Balakumar Sundaralingam and Karl Van Wyk for their help in evaluating MπNets and benchmarking it against STORM and Geometric Fabrics respectively; Ankur Handa, Chris Xie, Arsalan Mousavian, Daniel Gordon, and Aaron Walsman for their ideas on network architecture, 3D machine learning, and training; Nathan Ratliff and Chris Paxton for their help in shaping the idea early-on; Aditya Vamsikrishna, Rosario Scalise, Brian Hou, and Shohin Mukherjee for their help in exploring ideas for the expert pipeline, Jonathan Tremblay for his visualization expertise, Yu-Wei Chao and Yuxiang Yang for their help with using the Pybullet simulator, and Jennifer Mayer for editing the final paper.

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
