# OpenReview forum: "Motion Policy Networks"
_robot-learning.org/CoRL/2022/Conference — CoRL 2022 Poster_

### Official Review · Reviewer_3osu · 2022-07-23

**Originality:** Very Good
**Technical Quality:** Excellent
**Clarity Of Presentation:** Excellent
**Impact:** 4

**Recommendation:**

Strong Accept: I recommend accepting the paper and will argue for my recommendation even if other reviewers hold a different opinion.

**Summary:**

This paper primarily introduces a novel neural policy architecture - named M$\pi$nets - that can be used to generate robotic motions that achieve a particular goal configuration from a single depth image. To complement this, the authors create a novel extremely-large dataset of motion-planning problems and solutions on which they train their approach. Across extensive experiments, they show that M$\pi$nets not only outperform existing approaches in simulation, but also transfer well to novel tasks with noisy point clouds on real robotic hardware.

**Issues:**

**Questions**
(Note that these are not issues I have with the paper, but more of a curiosity/question. If possible, it would be great if you could include additional details answering this in either the main paper or supplementary materials.)
- Encoding the target in the point cloud: Is there some reason/intuition for why you decided to encode the target gripper position as part of the point cloud? It seems more straightforward to simply specify the target as a configuration ($q$) vector. Additionally, one of your ablations seems to show that if you remove $q_{t}^{||\cdot||}$ and instead supply this as part of the point cloud, performance degrades substantially. Wouldn't this suggest you can improve performance by also providing the target as an explicit vector instead of as part of the point cloud?



**Quality Of The Limitations Section:**

Limitations are addressed clearly

**Reviewer Expertise:**

3: The reviewer is fairly confident that the evaluation is correct

**Robotics Focus:**

Sufficient demonstration on hardware

**Strengths And Weaknesses:**

Strengths:
- Simplicity of the method: The presented M$\pi$nets architecture and loss functions used for training are rather simple, especially compared to certain baselines, yet perform remarkably well. This simplicity makes the method easy-to-understand and also provides good intuition for why it works.
- Extensive experimental evaluation: The authors' experimental evaluation is extremely thorough and extensive: there is not a single additional result I can think of that I would like to see.
- Well-written paper: The paper's writing and figures are extremely clear and easy-to-follow. Additionally, most important details are provided: I feel that a reader could re-implement the method after carefully studying the paper.

Weaknesses:
No major weaknesses I could think of.

**Summary Of Recommendation:**

Overall, I believe this is a strong paper that not only makes several useful contributions to the robotics community, but is also well-presented and easy to follow. In particular, the authors' method and novel dataset offer a new tool towards deploying motion-planning in real-world situations, which has been a goal of the robotics community for several decades. The experimental evaluation is thorough and convincing. I don't see any major flaws that could cause me to recommend rejection.

---

> ### Author Response · Authors · 2022-08-27
> **Response to reviewer 3osu**
>
> Thank you for your supportive comments. We greatly appreciate your enthusiasm and feedback. Here are our answers to your questions, but we will make these more clear in the paper.
>
> There are two parts to your question: 1) Why use target pose as opposed to configuration and 2) why not supply the target as a vector.
>
> 1. [This response was also shared with Reviewer S422] We feel that learning in task space is beneficial for several reasons. Long-term, we desire robots that make motions intuitive to human operators and partners, and humans prefer roughly straight line motion for the hand, as studied by https://link.springer.com/article/10.1007/BF00204593. Also, goals in manipulation are typically represented in end effector task space [https://arxiv.org/pdf/2106.01352, https://arxiv.org/pdf/2103.14127]. In a closed loop setting with a moving target, the traditional process of using IK to map task to configuration space can produce highly variable configurations, especially around obstacles.
>
> 2. If our intent were to learn a target in configuration space, we suspect that this numerical representation would be helpful. Indeed, this is what the authors of MPNets use. Our goal is to learn a target in end effector task space, and, we show in the Appendix, representing the 6D pose as an explicit vector hurt performance. We show this result in the Appendix. Prior work in the 6DOF pick-and-place literature, such as https://arxiv.org/abs/1905.10520 and https://arxiv.org/abs/1912.03628, have also demonstrated proficient results in encoding robot SE3 poses as part of the point cloud in PointNet-based network backbones.

---

> ### Author Response · Authors · 2022-08-27
> **Paper Revisions Relevant to Reviewer 3osu**
>
> In the revised paper, you will find a more thorough motivation for task space planning in Section 2 (Global Planning)

---

### Official Review · Reviewer_S422 · 2022-08-03

**Originality:** Good
**Technical Quality:** Very Good
**Clarity Of Presentation:** Good
**Impact:** 3

**Recommendation:**

Weak Reject: I recommend rejecting the paper, but will not argue for my recommendation if the majority of other reviewers have a different opinion.

**Summary:**

The paper proposes a neural motion planning learner, which takes a point cloud and a robot configuration as input and outputs the next configuration. The learned model can handle unknown, partially observable, and dynamic environments. Its outperforming performance over various baselines empirically proves its applicability to real-time practical scenarios.

**Issues:**

Major comments:
- Harder problems (narrow passages): As pointed out above, most of the problems shown in the paper and the video look relatively simple (wide passages in the configuration space). The only hard one is the limitation scenario in the video. Since the proposed framework is learning-based, its fast computation is not surprising compared to classical planning algorithms. I think a more rigorous analysis of whether challenging problems (e.g. arm going through multiple objects in between) can or cannot be solved is needed to understand how much completeness is sacrificed.
- Consistency (and task space): The authors emphasize the importance of the notion of consistency in many places, but its definition is vague and rather handwavy. It is defined in the paper as: "consistency describes how the motion would change in response to a perturbation in any dimension of the planning problem." Is it a subjective definition? Otherwise, please redefine this notion precisely. This confusion incurs sequential confusion as follows. I couldn't understand why learning in the task space is more beneficial than learning in the configuration space, which has some connection to consistency. Also, the motivation of hybrid planners is unclear. Since this is the primary motivation of the proposed framework, clarifying this point is critical.
- Being able to handle partial observability: The model is trained with fully-observable data, but it is unclear how the ability to handle partial observability at test time emerges.
- Hybrid planner: Because of the confusion about consistency, the fundamental difference between global planner and hybrid planner is vague, making the rest of the paper hard to understand. Moreover, in Section 5, what would be the fundamental differences between problems solvable by either one? Please clarify it. The word "hybrid" comes from the fact that it is a combination of the global planner and the local planner?


Minor comments:
- In the contribution bullet points, "46% better": The word "better" is not well-defined. Is it referring to efficiency or the success rate? Please clarify it.
- In the introduction, "enforce strong assumptions about ...": Please briefly explain the assumptions.
- In Section 2, global planning: The authors state that "optimal paths in configuration space can lead to unintuitive paths in a task space", which may be related to the notion of consistency. What would be intuitive paths in a task space? Distinguishing intuitive paths from unintuitive ones seems arbitrary and subjective. Please provide one example.
- In Section 3.3, "task spaces on the surface of the robot": I couldn't understand the relationship between the task space and the surface of the robot.
- In Section 3.4: Do the closed-loop rollouts mean that the loss is computed from not a single configuration but an entire trajectory?
- In Section 3.4, "We also sample novel scene point clouds on-the-fly during training": I couldn't understand this sentence. Does this mean one problem could have entirely different point clouds from the current scene? I am not sure how learning is possible from it.
- In Section 4.2, global planner: The authors state that the global planner fails if the smoother produces collisions. I am questionable if it is a major and fair cause of failure because it is just a wrong hyperparameter tuning.
- In Section 4.2, hybrid planner: Please clarify "create a consistent configuration space velocity profile", which is hard to understand.
- In Table 3, I couldn't understand why there is such a large gap between hybrid expert and global expert on the environment collision (i.e. 0.94 vs. 13.78), which is related to one of the above comments. Is it still happening even after a careful hyperparameter tuning of the global planner?
- In Section 5.1, the results of the hybrid planner: There are 11%, 2.06%, and 7.94% written in the text, but they don't correspond to any numbers in Table 3. Are they separate results?
- In Section 5.3: 16.41% depends on how the authors considered unknown regions of problems in training. Please explain the method briefly.
- Typo in Section 7: From "end-to-end neural molicy policies", remove "molicy".

**Quality Of The Limitations Section:**

Limitations are addressed clearly

**Reviewer Expertise:**

4: The reviewer is confident but not absolutely certain that the evaluation is correct

**Robotics Focus:**

Sufficient demonstration on hardware

**Strengths And Weaknesses:**

Strengths:
- The performance of the proposed framework is impressive, especially in dealing with dynamic environments.
- Evaluation is exhaustive as various metrics are used, and multiple baselines are compared.

Weaknesses (details are included in the Summary Of Recommendation and Issues sections):
- Relatively simple problems are evaluated. Problems having narrow passages, the most notorious challenge in motion planning, are not sufficiently assessed, in which a learning-based method is expected to work poorly.
- Some points (e.g. task space and the notion of consistency) made in the paper are vague, thus arguable.

**Summary Of Recommendation:**

The paper is well-organized, and its presentation is clear. Exhaustive evaluations are impressive, and the video demonstration clearly conveys the method's performance. I think the paper can further be improved by addressing the weakness points made above. In particular, whether to be able to handle more intricate problems having narrow passages is of significant concern.

---

> ### Author Response · Authors · 2022-08-21
> **Addressing Major Comments from S422**
>
> We thank the reviewer for their careful read of our paper and constructive comments. We will provide an updated PDF early next week, but meanwhile, here are our responses.
>
> ### Learning in Configuration Space
> We feel that learning in task space is beneficial for several reasons. Long-term, we desire robots that make motions intuitive to human operators and partners, and humans prefer roughly straight line motion for the hand, as studied by https://link.springer.com/article/10.1007/BF00204593. Also, goals in manipulation are typically represented in end effector task space [https://arxiv.org/pdf/2106.01352, https://arxiv.org/pdf/2103.14127]. In a closed loop setting with a moving target, the traditional process of using IK to map task to configuration space can produce highly variable configurations, especially around obstacles.
>
> ### Consistency and Hybrid Planner vs. Global Planner.
> We would like to apologize for our vague definition of consistency. We would like to further break down consistency into two metrics 1) expert quality and 2) repeatability of the planner. Repeatable input-output datasets are important for deep learning systems. Prior works have shown the deep learning systems deteriorate or require more data when using noisy labels [https://arxiv.org/abs/1512.06974, https://arxiv.org/abs/1511.02251].
>
> First, the quality of the expert affects policy learning. A common definition of expert quality in the behavior cloning/imitation literature is lower expert path length, which is correlated to traditional definitions of optimality (min cost) [https://arxiv.org/pdf/2108.03298.pdf]. Our task is to learn a policy in task space and the Hybrid Planner produces shorter task space paths–averaging 57cm ± 31cm, 95° ± 52°–compared to the Global Planner–averaging 61cm ± 39cm, 113° ± 55°–and hence is a better expert from that perspective.
>
> Second, in terms of repeatability, both the Global Planner and Hybrid Planner are sampling-based planners and do not produce repeatable paths by their very nature. Yet, the Hybrid Planner uses sampling to plan in a lower-dimensional state space (task-space is 6 dimensional) while the Global Planner samples in full configuration space (7 dimensions) hence the former has a smaller search space and will converge toward the optimal plan more quickly. The Hybrid Planner also employs Geometric Fabrics [Van Wyk et. al. 2021] to follow the task-space trajectory. Geometric Fabrics are deterministic, which further promotes repeatability in the final, configuration space trajectories. Essentially, this means that, if the sampling-based components of both planners were run twice for the same problem, we would expect the Hybrid Planner’s two solutions to be more similar to each other than the Global Planner’s solutions.
>
> In future work, we plan to dive into a theoretical explanation of consistency and connect it to our empirical observations of training MPiNets with different experts.
>
> ### Harder Problems
> Our intention with this paper is to demonstrate the power of this approach relative to its alternatives, mainly local policy methods. We agree with the reviewer that more challenging environments could be included, but our empirical tests demonstrate our environments are already challenging for existing local methods. For example, as shown in Table 7 in the appendix, STORM [CoRL’21] can solve 58% of hybrid-planner-solvable problems in the dresser setting, whereas Geometric Fabrics can solve 26%. Meanwhile, MPiNets can solve 97%. For dresser problems solvable by the global planner, each of these numbers drop by MPiNets’ success rate is nearly 55% higher than STORM.
>
> Our video examples were selected randomly and we apologize for not including more explicitly difficult problems. We have gone through our 1800-problem hybrid-solvable test set and picked out a set of 190 of the most challenging problems. We verified that these problems are more challenging than our overall dataset by using RRTConnect to find a feasible solution. For these harder problems, RRTConnect takes nearly twice as long to find a feasible solution, with an average 1.13 ± 3.51 seconds vs 0.64 ± 0.69 for the overall dataset. On these harder problems, MPiNets’ success rate is 95.24% vs 95.33% for the entire dataset. We have uploaded a video demonstrating MPiNets’ performance on one problem from each environment type. You can see these videos here https://bit.ly/3pzEKU8
>
> ### Partial Observability
> This quality–that the network can generalize to partial-view point clouds–is a well studied property of PointNet, which is a subcomponent of PointNet++ (the architecture we use). For example, in the original PointNet paper (https://arxiv.org/pdf/1612.00593.pdf, section 5.1, 3D Object Part Segmentation), they discuss a minimal drop in accuracy when using partial view point clouds for the task of part segmentation. We will include a more thorough discussion of this point in the final paper.

---

> > ### Author Response · Authors · 2022-08-21
> > **Addressing Minor Comments from S422**
> >
> > Thank you again for the helpful comments. We could not fit our responses into a single comment box and reply to your minor comments here.
> >
> > Minor Comments:
> > 1. “Better” should be “more successful.”
> > 2. The assumptions are explained in paragraph 3, but we will change this language to include examples in paragraph 1.
> > 3. Human motion preferences are discussed above. We consider this to be unintuitive because humans are not adept at reasoning in configuration space.
> > 4. These task spaces are a fixed set of points on the surface of the robot. We will amend the language to avoid this confusion.
> > 5. The loss is computed for single-step prediction. The closed-loop rollouts only refer to how we use the model during inference (i.e. online execution).
> > 6. During training, we generate a new point cloud for every training example. Each training epoch, the network is exposed to 163.5 million (3.27 million trajectories * 50 time steps per trajectory) previously unseen point clouds.
> > 7. We do not intend to claim that these failures are insurmountable, but instead that they are limitations of the off-the-shelf solutions when generating a large dataset. If the Global Planner takes 16 seconds (as is average as reported in Table 1), it takes over 3.3 years of computation time to generate our global planner dataset. We will amend this language to emphasize that these are not algorithmic failures, but limitations we faced during scaling.
> > 8. After planning an end effector path, the Hybrid Planner relies on Geometric Fabrics to move between each waypoint along the end effector path. However, the Geometric Fabrics sometimes fall into local minima between waypoints, leading to very slow movements along the path. After stitching all of these motions together, the overall trajectory can appear to stop-and-start as the waypoints change. We use the smoothing step to retime the entire trajectory to appear as a single smooth motion through the waypoints. See https://bit.ly/3c5ORwG for a video example of the hybrid planning process.
> > 9. These numbers are of MPiNets performance trained with different expert datasets. As described above, our assessment is that the difference in performance can be attributed to expert quality and our hyperparameters were chosen carefully according to our scaling needs.
> > 10. These numbers have a typo, but should appear in Table 2 in the Appendix. We apologize for the confusion, will correct them in the amended PDF, and will direct the reader to the appendix.
> > 11. We use an 3D occupancy grid and consider a space occupied if a point is registered there. We do not fill any unobserved spaces. We will add further detail here as well.
> > 12. We will fix this!

---

> > ### Comment · Reviewer_S422 · 2022-08-26
> > **Follow-up question on learning in configuration space**
> >
> > Thank you for carefully addressing my comments and additional evaluations. Most of my questions are clarified, and I have one follow-up question, which I think is critical.
> >
> > The authors claim that learning in task space may be more beneficial than learning in configuration space. I agree with the claim from the human interpretability perspective, but I am not sure from a learning perspective. In fact, my intuition says it's the opposite because a task space can be obtained through complex forward kinematics (the higher robot DOFs, the more complex forward kinematics). Thus, learning directly in configuration space brings less burden to the neural network by skipping additional transformation. Because of this, the paper's method may handle geometrically challenging (i.e. narrow passage) problems worse than the counterpart, so I found the paper's claim counterintuitive and surprising. Can I consider this work as an empirical evidence of the claim, or am I overthinking?
> >
> > I see that new simulations show more complex problems than the original submission. However, unless we take a close look at narrow passages in the configuration space, it is hard to say whether those problems are really hard from a motion planning perspective. Thus, I do not think this sufficiently justifies the above my point. Do the authors have any thoughts on this question?

---

> > > ### Author Response · Authors · 2022-08-27
> > > **Follow up to S422 Regarding Configuration Space and Hard Problems**
> > >
> > > Thank you for your response and further discussion. If you have further questions or concerns on these topics, please comment and we will reply promptly.
> > >
> > > ### Question 1
> > > - We do not make any claims as to whether it is easier or harder to learn with configuration space targets.
> > > - We chose task space not for its ease-of-learning, but for our intended use-case. As we discussed in our previous response, our aim is to create a policy that is useful for manipulation and intuitive for a human partner or operator.
> > > - Whether the targets are specified in task or configuration space, the network must learn an internal kinematic representation to avoid obstacles. Obstacles are represented in the point cloud, which is in Euclidean space, whereas the robot is represented in configuration space.
> > > - While we did not test configuration space targets, we did test using MSE in configuration-space as the loss (instead of the task-space behavior cloning losses). Overall performance was similar between the two loss representations, but the MSE loss had approximately double the collision rate. This is detailed in the appendix.
> > >
> > > ### Question 2
> > > - In terms of difficulty, we chose this environment because MotionBenchMaker [https://arxiv.org/abs/2112.06402, RAL 2021] is an accepted benchmarking suite for robotic manipulation tasks. And, as demonstrated in their paper, this is the most challenging among their environments for a single-arm manipulator. By challenging, we mean that empirically, these problems take the longest for a sampling based planner to find a feasible solution. As mentioned before, we verified this empirically on our end as well by comparing the planning time to our other environments and problems therein. We do not have a way to determine whether the inverse mapping of the narrow Euclidean geometry produces a narrow passage in 7D configuration space, other than to examine its empirical planning times when using a uniform sampler in configuration space (which we did using RRTConnect). If you have a suggestion on how to rigorously measure this, please comment and we are happy to perform this evaluation.

---

> ### Author Response · Authors · 2022-08-26
> **MPiNets in a More Difficult Environment**
>
> In addition to our previous comments on a more difficult subset of problems, we generated and trained a model using a randomized cage-like environment. This class of environment comes from MotionBenchMaker, a recently published benchmark set for robotic manipulation [https://arxiv.org/abs/2112.06402, RAL 2021], in which the authors demonstrate that the cage is the most challenging amongst their environments for a single arm manipulator. We generated data using their same randomized procedural generation, which randomly positions and orients the cage, as well as chooses a randomized task-oriented pose within the cage. To compare difficulty with our dataset, we constructed a set of 200 cage problems solvable by our Global Planner, where the robot reaches from a neutral configuration into the cage or visa-versa (100 of each). When running RRTConnect, the average time to find a solution was 13.10 ± 12.77 seconds, over 20 times longer than that of our dataset.
>
> In these environments, we were unable to generate a large dataset of Hybrid Planner solutions, largely due to local minima preventing Geometric Fabrics from reaching the target. Instead, we generated 200K demonstrations with the Global Planner. No environments or problems are shared between the train and test sets. After training MPiNets from scratch on this data for 4.5 days, we observe an 82% success rate on the test set with a 15.5% collision rate. We also used this data to fine-tune the model trained presented in the paper, i.e the one trained with 3.27M Hybrid Planner demonstrations. This fine-tuned model reached 85.5% success with 11.5% collision rate after training for 17 hours. These numbers are in-line with MPiNets trained with the Global Planning Expert and evaluated on the Global Planner-solvable test set (75.06%). We suspect the improved performance in the cage setting is due to the smaller variation amongst the environments. With more time, we expect that we could generate Hybrid Planner demonstrations and observe similar performance to what we demonstrated on our environments in the paper.
>
> See here for a video demonstration of the fine-tuned policy in the cage settings: https://bit.ly/3TmvD6H

---

> ### Author Response · Authors · 2022-08-27
> **Paper Revisions Relevant to Reviewer S422**
>
> In the updated paper, you will find a discussion on consistency in Appendix B.
>
> We have updated our motivation for task-space planning in Section 2 (Global Planning).
>
> We have corrected your "minor" comments as well (the section numbers have not changed)

---

### Official Review · Reviewer_V556 · 2022-08-06

**Originality:** Very Good
**Technical Quality:** Very Good
**Clarity Of Presentation:** Excellent
**Impact:** 4

**Recommendation:**

Strong Accept: I recommend accepting the paper and will argue for my recommendation even if other reviewers hold a different opinion.

**Summary:**

This paper presents an end-to-end neural motion policy network that learns to generate collision-free trajectories from a point cloud of the scene, the goal state and current joint configuration. It is trained on a large-scale dataset of millions of sample motion planning problems from hundreds of thousands environments. The approach, referred to as M$\pi$Nets, is directly comparable to the MPNets [11] algorithm, showcasing improved performance when sufficient data is available without requiring collision-checking and being able to react in dynamic environments. The data used to train an M$\pi$Net is generated from an expert hybrid planner that leverages the strengths of global and local planners.

**Issues:**

Comments listed in the weaknesses section should be addressed.

**Quality Of The Limitations Section:**

Limitations are addressed clearly

**Reviewer Expertise:**

5: The reviewer is absolutely certain that the evaluation is correct and very familiar with the relevant literature

**Robotics Focus:**

Sufficient demonstration on hardware

**Strengths And Weaknesses:**

**Strengths**

- The paper is well-written and enjoyable to read.
- It proposes an interesting solution to difficult motion planning.
- The fact that it is trained on raw point cloud observations makes the approach amenable to sim2real without any additional modifications.
- The evaluation of accuracy and performance of M$\pi$Nets vs. baselines and other SOTA is sufficient and rigorous (except for computation times – discusses in weaknesses)
- Simulated experiments are impressive.
- Inclusion of failures in the video is appreciated.

**Weaknesses**
- While the comparisons and metrics used to evaluate M$\pi$Nets do a very good job at analyzing its performance based on task/collision-avoidance success, there is no mention about computation time either at training or test time. Only until we go to the appendix we see that M$\pi$Net takes 1 weeks to train with 8 NVIDIA Tesla GPUs. This seems like a very important detail to leave out from the main paper. Furthermore, I would have expected to see a comparison of training/testing time vs. MPNets. How much compute is needed to run the network at test time? Do you need the 8 NVIDIA Tesla GPUs? When proposing an algorithm that requires very heavy computation, one must consider that this might not be accessible or even feasible for certain scenarios or circumstances – hence, the hardware requirements must be transparent.
- Throughout the text it is noted that low performance due to distribution shift could be alleviated by combining the global and local planners. It would be great to see in the paper an explanation as to why this is difficult. Couldn’t you combine the successful demonstrations from both the global and local experts? I assume this might introduce inconsistencies in the data and instead of improving the overall performance it would deteriorate it. Is this the case? It would be great to include such comparison or some extended analysis on this and some suggestions as to how these experts could be combined. Have the authors thought of employing an expert combination strategy like Bayesian Model Combination as done here for an RL setting: https://dl.acm.org/doi/10.5555/3327546.3327623
- In Table 2, the success rate for M$\pi$Nets tested on the solvable problems seems extremely low 5.5%, could this result be elaborated? What is so fundamentally different between the Hybrid Expert datasets and the MPNets-Styles?
- The two control methods should be mentioned in the main text (frequency for closed-loop control) and further details remaining in appendix.
- It seems that the points clouds are always uniformly sampled both at training and test time? Is this correct? What is the performance with truly noisy input point clouds?
- The catastrophic and erratic behaviors that are due to distribution shift make it difficult to see this learned end-to-end network working on real-world scenarios. Apart from using/generating more data or allowing corrections to the learned model like DAGGER, there must be a fundamental safety component that allows the robot to at minimum not collide with the environment. Have the authors thought about introducing such a mechanism to the framework?  It would be interesting to see this discussion.


**Further Improvements/Suggestions**

- To learn a robust policy that is “consistent” with the expert demonstrations the authors might want to consider augmenting the imitation loss with some notions of stability as in:
https://arxiv.org/pdf/2205.14812.pdf
- It would be interesting to include more metrics to evaluate the distinct behaviors of each motion planner, like: obstacle clearance, near-obstacle velocity and motion legibility.



**Summary Of Recommendation:**

This paper presents an interesting solution to a classical problem in robots - efficient collision free motion planning. This is done by an end–to-end motion policy network that takes as input a point cloud of the scene, the target and the joint configuration of the robot and is learned by a large scale dataset of expert demonstrations provided by planners. While the idea of learning from expert planners is not novel, the performance and range of environments that this motion policy network can handle is unprecedented. The paper needs to be more forthcoming about the computation power necessary to train and run such networks.

---

> ### Author Response · Authors · 2022-08-27
> **Response to Reviewer V556**
>
>
> Thank you for your time and careful consideration. And thank you for your helpful suggestions–these will be invaluable as we move forward in this research. We will address your comments in our updated draft, but here are our responses:
>
>  1.  We answer this questions in two parts:
> 	 1. The 8 GPUs are only required for training. During inference, we used single-GPU machines. We used different hardware for our simulated trials and real robot demonstrations. For our simulated experiments, we use a desktop with CPU Intel(R) Core(TM) i9-9820X CPU @ 3.30GHz, GPU NVIDIA A6000, and 64GB of RAM. For our hardware demonstrations, we used a desktop with CPU Intel(R) Core(TM) i7-7800X CPU @ 3.50GHz, GPU NVIDIA Titan RTX, and 32GB of RAM. We will add the computational requirements to the limitations section in the Appendix.
> 	 2. MPNets converges much more quickly–using the same training setup (and the same data), the MPNets model converged in 15 hours.
> 2. We will add an experiment trained with a dataset combining both planners (half of each, covering non-overlapping problems). We observed an averaging behavior between the models. When trained on a combination, both target convergence and collision rates lay in between the models trained entirely with one expert or the other. Compared to the hybrid-planner model, target convergence was especially higher on the test set of problems solvable by the global planner. We attribute this to data coverage because the hybrid planner is not always effective at reaching targets. For collision rate, as you mention, we assume that the deteriorating performance is due to inconsistencies in motion strategy compared to the Hybrid Planner data alone.
> 3. The MPNets data is 10000 examples in a tabletop setting with mostly top-down task oriented poses. The Hybrid Planner dataset includes all three classes of environments with a much wider distribution of obstacles and target poses. We attribute this poor performance to two things: 1) Lack of coverage in the training data 2) too little data overall. Meanwhile, MPNets uses a more traditional approach (with a collision checker), which makes it better able to adapt to out-of-distribution problems.
> 4. We will move this into the main paper. Thank you for the suggestion.
> 5. This is correct. Point clouds are sampled uniformly during training and most of our tests, although we also demonstrate performance using point clouds sampled from a depth image. We will add an experiment to the Appendix detailing performance when adding $0$-meaned Gaussian noise to the point clouds with varying standard deviations ($\sigma$). The network's performance is roughly constant with noise increasing up to $\sigma=3cm$ , when success drops to 89.28%. You can see the decaying performance with increased noise in this graph https://bit.ly/3dMFgva.
> 6. We agree with this assessment and it is in our plan for future work. We will add this to the discussion.
>
> In regards to your suggestions for further improvements, we will investigate these in future work. In particular, we will look into bayesian model combination and stability loss. As far as metrics, we are actively developing what we hope will be a canonical set of metrics for learned motion control in manipulation. We thank the reviewer for these suggestions and will certainly consider them moving forward.

---

> ### Author Response · Authors · 2022-08-27
> **Paper Revisions Relevant to Reviewer V556**
>
> In the updated paper, you will find information on inference hardware in Appendix F.
>
> We have added the experiment combining planners to Appendix J.
>
> We have amended our language to be more clear about the performance comparison to MPNets in section 5.1.
>
> We have added a description of the two control types to section 5.6.
>
> We have added the experiment with noisy point clouds to Appendix J.
>
> We included a brief discussion on using an external safety system in limitations (section 6).

---

### Official Review · Reviewer_Eji4 · 2022-08-06

**Originality:** Good
**Technical Quality:** Fair
**Clarity Of Presentation:** Very Good
**Impact:** 2

**Recommendation:**

Strong Reject: I recommend rejecting the paper and will argue for my recommendation even if other reviewers hold a different opinion.

**Summary:**

The paper presents an end-to-end deep learning based motion planner that is designed for use with manipulator arms and point cloud representations of obstacles. The goal of the work is to increase planning speed versus previous methods. For example, as compared to sampling based motion planning, graph search, and other forms of optimization. The algorithm is trained using millions of expert trajectories and two expert systems (one based on a sampling based motion planing algorithm, and another based on a hybrid algorithm). Extensive experiments and ablation studies are performed in simulation and there is also a statement that the work has been evaluated on a real robot.




**Issues:**

1) Right now the lack of information regarding how training and test sets related makes it hard to understand how to apply this method because there is not evidence of how much a trained model is able to generalize about what it has learned. For example, does training on a table top produce a model that can be used to plan on a counter or on the floor? How close in shape do the objects in the test cases need to be to the objects used in the training examples? Adding such discussion would improve the paper.  Even better would be the addition of empirical assessment of these things.

2) Deployment in dynamic environments is cited as a motivation for the work multiple times. However, the paper does not appear to contain the results of experiments from dynamic environments. Performing such experiments and reporting how performance changes (e.g., success rate) as a function of, e.g., the speed of environmental changes would improve the paper. Otherwise, claims of deployment in in a dynamic environment should be removed from the paper.

3) Considering cases where solutions do not exist is just as important as considering cases where solutions do exist. Testing what happens when no solution exists will improve the paper (how often is there a false positive path result that leads to a collision?).

4) The discussion of sampling based motion planning algorithms contains many statements that are not true. See (A-C) in the "Strengths and weaknesses" section.  This language should be changed to include more nuanced discussion of A-C.

5) The paper states that "MπNets is only trained for single-step prediction, but we use it for closed-loop rollout." Yet, the paper also states that Geometric Fabrics [9] and STORM [10] that "deploy reactive local policies and assume that local decisions will lead to globally acceptable paths" but have worse performance in practice. All three (MπNets, Geometric Fabrics, and STORM) appear to be making local decisions (or single-step prediction) and have an assumption that stitching them into global paths will work. The paper can be improved by adding additional information/discussion about why performance differs between these methods

6) Despite claims of sim-to-real transfer, the hardware experiments are anecdotal and not empirical. Adding empirical hardware experiments would improve the paper by quantitatively demonstrating the sim-to-real transfer. This would also help to alleviate worries of training vs. test sets (that is, if the training set is from simulation and the test set is from real hardware).

7) The  quality or generality of the method with respect to training set size and scope should be discussed in the limitations sections. For example, how close does the training and test set need to be to get decent performance in practice. How  well does training on simulation data generalize to testing on hardware?


**Quality Of The Limitations Section:**

Additional details required

**Reviewer Expertise:**

4: The reviewer is confident but not absolutely certain that the evaluation is correct

**Robotics Focus:**

Highly relevant to robotics but no hardware experiments

**Strengths And Weaknesses:**

The main strengths of the paper are as follows:
1) The training set is very large "nearly 300x larger than previous work" according to the paper.

2) Many different metrics were collected including: success rate, time, rollout target error, collision rat, and smoothness.

3) The number of simulations performed is extensive.

4) The experiments show that using the learned model instead of planning from scratch reduces runtime and increase smoothness (though the model also has decreased performance with respect to other metrics such as success rate).


The main weaknesses of the paper are as follows:
1) There is little discussion of the relationship between the training and the test sets. Therefore, it is impossible for this reviewer to evaluate if the test set was chosen fairly, and hence, the quality or generality of the experimental results cannot be assessed.

2) There does not appear to be experimental data/results regarding the performance of the algorithm in dynamic scenarios, despite this being cited as a benefit of the proposed method in the paper, nor when a solution is asked to solve a problem in which a solution does not exist. Testing in cases that do not have solutions is as important as testing in cases that do have solutions.

3) The discussion of graph search and sampling based motion planning algorithms contains many statements that are not true (see A-C below).

4) Despite claims of sim-to-real transfer, the hardware experiments are anecdotal and not empirical.

5) The  quality or generality of the method with respect to training set size and scope should be discussed in the limitations sections. For example, how close does the training and test set need to be to get decent performance in practice. How  well does training on simulation generalize to testing on hardware?


-- incorrect claims about sampling based motion planning and graph search --

A) The paper states, "Global planners such as RRT [1] optimize for speed and completeness but fall short on optimality." This quote communicates a fundamental misunderstanding about the goals of feasible sampling based motion planning. RRT and other *feasible* sampling based motion planning algorithms are designed to answer the question: "is it possible to reach the goal from the start?" Answering this question in the affirmative is possible given any valid path, and so all valid paths are considered equally "good" by RRT.  So, saying feasible algorithms  "fall short" with respect to optimal is incorrect.  Using RRT when one cares about optimality is simply using the wrong tool for the job.

B) The paper states, "Search-based planning algorithms ... are slow but find least-cost paths... Sampling based planners are typically much faster..." Whether either class of algorithms is "fast" or "slow" depends on how they are used. One can trade speed for better resolution. Moreover, graph search on n nodes is typically many orders of magnitude faster than using sampling based motion planning to create a graph of n nodes from scratch. So, if we fix a lattice graph of a certain resolution then graph-search will require a runtime that scales to the power of the number of C-Space dimensions; on the other hand, if we fix the number of nodes and neighbors then the runtime will remain constant vs. the number of dimensions but the resolution (of nodes per unit hyper-volume) will decrease.

C) The paper states, "The Global Planner {AIT*} is theoretically complete and optimal, but in practice, we observed several common failures types" and "Global Planner is theoretically complete." This is untrue. AIT* is neither complete nor optimal.  Rather, AIT*, as with all almost-surely asymptotically optimal sampling-based planing algorithms, is *probabilistically complete* and *asymptotically optimal*. Inherent in these definitions are the facts that more planning time will yield better results, on average, and that there is a stochastic nature to the algorithms' performance. It may also be the case that no valid solution exits (in which case the algorithm will not return a solution).



**Summary Of Recommendation:**

The idea is interesting and within the scope of the conference, yet there are also a number of errors in the paper regarding previous work, missing details regarding the training and testing process, and claims that are not backed up by experimental results. The main contribution of this paper is the proof of concept demonstration in simulation that end-to-end manipulator planning is possible given point cloud data. While many experiments are run, it is hard to assess the results because there is no indication of how closely the test set resembles the training set. There do not appear to be any empirical results from experiments run in dynamic environments, nor in cases where solutions do not exist, nor on robotic hardware. While the core idea is interesting and a good goal, this version of the paper leaves too much room for improvement.

---

> ### Author Response · Authors · 2022-08-22
> **Response to Comments from Reviewer Eji4, Part 1**
>
> We thank the reviewer for their constructive comments and feedback. We will try to answer them in order. We will upload an updated version of the paper early this week incorporating these discussions.
>
> ### Relationship between the training and test sets
> We apologize for not including more detail here. We use the same randomized procedural generation to generate planning problems (environment, start, goal) in both train and test. While they are drawn from the same distribution, the test set’s environments, starts, and goals are all unique from those in the training data. To make the distribution transparent, we will detail the precise parameter ranges to the appendix, as well as open source the data generation code upon publication. We will also include a more informative figure demonstrating the distribution of environments in the main body of the paper (https://bit.ly/3c7GGQD).
>
> We will also add an experiment to the appendix where we replace tabletop objects with randomly sampled, yet similarly sized, YCB objects to demonstrate robustness to point cloud shape. In these experiments, MPiNets’ performance drops from 94.67% to 88.33%. Here is a video example of these point clouds and the generated trajectories: https://bit.ly/3QK5k8R. In future work, we believe that we could further improve our performance on novel tabletop clutter by diversifying our object assets during training such as the following prior work: https://arxiv.org/abs/1912.03628.
>
> ### Dynamic environments
> We do not claim that MPiNets can operate in every dynamic scene, merely that its reactive nature as a policy by design allows it to react to a changing environment without recomputing an entire path. This naturally falls out of the architecture and we show videos of this behavior to emphasize the properties of our end-to-end system (no recomputation of a perception model is needed). This is an important benefit of using a reactive policy over a planner. Furthermore, there has been several prior work using instantaneous visual policies to react to dynamic scenes, e.g. https://arxiv.org/abs/1504.00702.
>
> Our key contribution is MPiNets’ performance ability to handle such a wide variety of environments, as demonstrated by our stationary experiments through their sheer number and randomness. That is, when a static snapshot of the scene is within distribution, our policy can choose a reasonable action. Therefore, the policy will be able to react to changing environments as well, assuming the changes are not so fast as to invalidate the actions before the next inference step. Demonstrating the trade-off between computation speed and environment speed speaks as much to our specific hardware as the network architecture.
>
> To further illustrate the trade-off with environment speed, we will provide an experiment in the appendix demonstrating how the network’s success diminishes with higher speed dynamic objects. You can see a video of one such trial here: https://bit.ly/3caj6CJ. We tested 1000 problems at four different speeds. All problems use the same table, but each has a random starting configuration, target, and initial block location. The block is then moved along the table at varying speeds. All problems are solvable when the block is static and we test three speeds: low, medium, and high. For the static, slow, medium, and fast speeds, the success numbers are 100%, 88.1%, 57.4%, and 28.3%, respectively.
>
> However, we were cautious to include this experiment in the original draft because the space of dynamic, 3D manipulation problems, even amongst our settings, is very large and thus hard to cover thoroughly. Alwala et. al. (https://arxiv.org/abs/2011.07171) evaluate in an environment with randomly moving cubes through space, but this setting is neither realistic nor physically possible. Creating a fair, representative set of realistic dynamic environments is a research problem itself and beyond the scope of this paper.
>
> ### Erroneous Planning Statements
> Thank you for pointing these out. We are sorry for mischaracterizing these algorithms and will correct our statements immediately. Our goal in the introduction (and related work) was to emphasize exactly your point regarding the utility of different planners, their trade-offs, and how to choose the “right tool for the job.” Our aim in the description of our global planning pipeline was to describe the practical limitations of off-the-shelf tools we faced when designing a system at such scale, not the limitations of the theoretical algorithms.

---

> > ### Author Response · Authors · 2022-08-22
> > **Response to Comments from Reviewer Eji4, Part 2**
> >
> > Thank you again for your comments. We want to reply to each thoroughly and weren't able to fit all of our responses in a single comment box.
> >
> > ### Local Controllers
> > We discuss this briefly at the end of 5.1–Local Task Space Controllers, but will add a more thorough explanation to the appendix. The difference in behavior stems from the fact that the other local methods do not have any experience in these environments and therefore make local decisions uninformed by long-term consequences. Meanwhile, the network has learned a general pattern of behavior based on environment observations. With prior knowledge of a specific environment, the other methods could be tuned or augmented with waypoints to exhibit better behavior.
> >
> > ### Negative Test Cases.
> > We apologize for not including this and will add it to the revision. We have constructed some impossible problems where the target pose is in collision and will add these experiments to the appendix. The policy collides in 63.5% of these problems. This limitation is to be expected as the network is not trained to handle these circumstances. We will emphasize this in the limitations section. In future work, we intend to focus on a safety mechanism to guard against invalid problems.
> >
> > ### Empirical Hardware Trials
> > Our goal in the evaluations was to isolate our network architecture from other system components and fairly evaluate its performance. In a physical system, there are many confounding variables–the choice of camera calibration and segmentation algorithms, low-level controllers and smoothers, and system tuning. We do not make claims that this is ready for immediate deployment in a production environment. In section 3.5, we state that we have run “qualitative experiments," but we will amend the language to make it clear that these are demonstrations, rather than rigorous trials. The purpose of these demonstrations is to illustrate how this technique can transfer to hardware, not that our system implementation is ideal.
> >
> > ### Improving Limitations Section
> > We mention the in-distribution requirement in the limitations (second sentence) and further discuss in the limitations section in the Appendix (the first paragraph). But, we will further explain the ways in which the data can be out-of-distribution in the appendix. We will also reiterate the training data size requirement shown in Figure 4.

---

> ### Author Response · Authors · 2022-08-27
> **Paper Additions Relevant to Reviewer Eji4**
>
> In the updated paper draft, you will also find a thorough description of the environment parameters in D, as well as a more descriptive statement on the relationship between train and test in section 5. We have included the results to the YCB experiment, as well as adding noise to the point clouds, in Appendix J. And, we have discussed the limitations of the train-test relationship in the Section 6 (limitations).
>
> The description of the dynamic environment experiment is in Section 5.5.
>
> We have also corrected our statements about search and sampling based planners in the introduction and related work, as well as included more accurate characterizations of the limitations of our global planning pipeline in Section 4.2.
>
> Our description of the performance with regard to impossible problems is in Appendix J.
>
> We have improved our statements on our real robot demonstrations in section 5.6
>
> We have also improved the limitations sections in both the main paper and the appendix to make them more thorough.

---

### Official Review · Reviewer_oeVp · 2022-08-09

**Originality:** Good
**Technical Quality:** Good
**Clarity Of Presentation:** Good
**Impact:** 2

**Recommendation:**

Weak Reject: I recommend rejecting the paper, but will not argue for my recommendation if the majority of other reviewers have a different opinion.

**Summary:**

The authors propose Motion Policy Networks, a learned policy trained on data from classical motion planning solutions. The policy takes as input a configuration space state and a point cloud and outputs configuration space actions. The authors propose additional losses based on motion planning and collect data from motion planning solutions. M\piNets then train a policy and roll out the actions for solve planning problems. This approach is ablated over each loss, data source, and compared to several motion planning techniques and shown to scale well with data and achieve high performance.

**Issues:**

See above, primarily with respect to framing.

**Quality Of The Limitations Section:**

Limitations are addressed clearly

**Reviewer Expertise:**

3: The reviewer is fairly confident that the evaluation is correct

**Robotics Focus:**

Sufficient demonstration on hardware

**Strengths And Weaknesses:**

The paper is clear and the experiments are well done. The point cloud representation is nice and general, especially compared to representations used in many motion planning and learning + motion planning algorithms. The experimental environments are nice and reasonably complex. The figures are clear as well. I think Figure 3 is informative and clear, that policies have more performance as data increases, whereas structured approaches like MP-Nets outperform in a limited data regime. The ablations are well done and thorough (ablating the loss, data sources, observation, etc) – and informative.

The primary weakness is the framing of the approach. This approach at its core is behavioral cloning in configuration space with additional loses and data training sources. But the authors discuss in detail the motion planning literation as well as learning for motion planning, and ablate and compare to baselines in those fields thoroughly. E.g., they compare to MP-net, which they rightly note requires collision checking, it also has guarantees of falling back on motion planning.

Instead the authors should be comparing to baselines for learned robotic policies and should be framing the paper accordingly. In framing this way, the authors can discuss data sources more clearly. Questions such as BC data if the data is smooth, noisy, has global coverage are interesting and useful for learning policies. The authors can also show how task-specific losses for motion planning help and hurt.

Minor comments:
- Can you clarify again why the global planner fails on hybrid problems, when I thought hybrid problems require a global solution to then be smoothed.
- The authors should show an image of the planning problems in the main body of the paper.



**Summary Of Recommendation:**

The authors have written a thorough and clear paper, but it is framed around motion planning when the algorithm is an imitation learned policy with ablations over data sources and task-specific losses. The experiments are thorough and indeed an interesting comparison with motion planning, but it would be useful to better understand how it fits within the imitation learning literature. It would also make comparisons for when learned policies outperform motion planning and when they can learn from motion planning more clear.

---

> ### Author Response · Authors · 2022-08-20
> **Response to Reviewer oeVp: behavior cloning, task losses, and off-the-shelf global planner limitations**
>
> We thank the reviewer for their time and considerate comments. We will update the paper will comments from all reviewers early next week, but in the meantime, we will respond to each of the major questions and comments here:
>
> ### Framing of the Paper:
> Regarding the primary concern of the work, we apologize for not including a more thorough discussion of the behavior cloning literature in the related work and will do so in a final version of the paper. We agree that our system is fundamentally just a form of behavior cloning or supervised learning. As far as we are aware, there is no behavior cloning technique or model architecture that has been able to successfully learn such complex motion planning tasks in the 6-dof arm manipulation domain, in unknown scenes, using a point cloud as the scene representation as we have shown with MPiNets.
>
> We will discuss other papers in the behavior cloning literature in the related work. In lower-dimensional problems, such as autonomous driving, work such as https://arxiv.org/pdf/1904.08980.pdf have demonstrated the challenges of behavior cloning due to distribution shift, long-tail behavior, and variance. In the manipulation domain, Mandlekar et. al. CoRL 2021 https://arxiv.org/pdf/2108.03298.pdf demonstrated how behavior cloning outperforms batch RL and other learning methods on a benchmark suite of manipulation tasks and datasets.
>
> In our work, we demonstrate that many of these challenges with behavior cloning can be overcome with enough data from a suitable expert in conjunction with data augmentation strategies. In our evaluations, we use randomly generated problems to demonstrate our robustness to variance and perform long rollouts to demonstrate our robustness to distribution shift over time. Since the primary domain and task of this work is on the motion planning problems, we compare against planners, analytical controllers and MPNets (Qureshi et. al. 2019) simply because these are the state-of-the-art for robot motion generation.
>
> ### The impact of task-specific losses
> The reviewer asked “how task-specific losses can help and hurt”. In our work, we focus only on the task of motion planning in unknown scenes from point cloud observations. We show in Section I of the Appendix how our task-specific geometric behavior cloning loss results in significantly lesser collisions (0.89%) as opposed to using the standard Mean Squared Error behavior cloning loss (2.39%). We also show that when trained without the collision loss, our policy collides more often–2.11% vs 0.89% when trained with the collision loss.
>
> ### Global Planner Failures on Hybrid problems
> We attempted to address this question in Section 4.2, but will clarify further in the final version. What we call the “Global Planner” is a combination of a probabilistically complete planning algorithm and a collision-aware, spline-based smoother. The smoother is an essential component because the planner’s raw output path is piecewise linear in configuration space. To deploy on a real-robot, we desire a smoothly changing velocity profile for the entire trajectory, which is not possible for a piecewise-linear trajectory without some smoothing. The “Global Planner” fails on these problems either because the planning algorithm times out or the collision-checking resolution for the smoother is too coarse and produces a trajectory that, when executed in simulation, produces a collision. At present, the majority of these collisions arise from these false-positives during smoothing. While we could have used a finer resolution for collision checking during smoothing, we had to consider trade-offs between planning time, accuracy, and our compute budget, especially given the size of our dataset. For evaluation, we chose to use the same parameters to demonstrate the successes and failures of the planning implementation we used for data generation.
>
> ### Image of the Motion Planning Problems
> We will include an image with planning problems in the main paper, as well as a more comprehensive visualization in the video. Here is a link to the image we will include: https://bit.ly/3PBH9bu

---

> > ### Author Response · Authors · 2022-08-27
> > **Paper Additions Relevant to Reviewer oeVp**
> >
> > In the updated paper draft, you will find more discussion on imitation learning and behavior cloning in the related work. We have also added more description on the limitations of our Global Planning pipeline in Section 4.2

---

### Author Response · Authors · 2022-08-27
**Updated Paper Draft**

We would like to thank the reviewers again for your time and energy. We have added additional detail to the paper according to your comments and attached it here. All modified sections are marked in red to make them easier to find. We have also included the Appendix in this document for ease of reading.

---

### Meta-Review · Area_Chair_a7yn · 2022-08-10

**Recommendation:** Accept (Poster)
**Confidence:** 4

**Metareview:**

Summary.
Motion Policy Networks, a learned policy trained on data from classical motion planning solutions, are proposed by the authors. The authors propose additional losses and collect motion planning data. MpiNets implements a policy to solve planning problems. This method scales well with data and performs well when compared to other motion planning strategies.
The research describes a deep learning-based motion planner for manipulator arms and obstacle point clouds. The project aims to speed up planning. Comparing sampling-based motion planning, graph search, and other optimizations. Training the algorithm uses millions of expert trajectories and two expert systems (one based on a sampling based motion planing algorithm, and another based on a hybrid algorithm). Extensive simulation experiments and ablation research have been evaluated on a real robot.
This research provides an end-to-end neural motion policy network that generates collision-free trajectories using a scene point cloud, objective state, and joint configuration. It's trained on millions of motion planning issues from hundreds of thousands of situations. The MNets technique is comparable to the MPNets [11] algorithm, displaying greater efficiency without collision-checking and the capacity to react in dynamic settings


Strengths
1. Well-written study with well-executed experiments.
2. The point cloud representation is excellent and broad compared to various motion planning and learning+motion planning techniques. Attractive and challenging experimental situations.
3. The training set is "300x greater than previous efforts"
4. Success rate, time, rollout target error, collision rate, and smoothness were measured.
5. Experiments demonstrate that utilizing the learnt model decreases runtime and increases smoothness.
6. It proposes a clever motion planning solution.
7.  Because it's trained on raw point cloud data, sim2real doesn't need any adjustments.
8. MNets' accuracy and performance are evaluated against baselines and other SOTA.
9. The suggested framework excels in dynamic contexts;
10. The given Mnets design and training loss functions are simple, yet work well. This simplifies the process and explains why it works.

Weaknesses

1. The training-test set connection is unclear and it cannot be ascertained if the test set was chosen fairly, thus the quality or generalizability of the experimental results cannot be easily evaluated.
2. There are no experimental data/results on the algorithm's performance in dynamic circumstances, despite this being a benefit of the proposed method.
3. Graph search and sampling-based motion planning methods contain many erroneous statements as the reviewers have noted.
4. Hardware trials are anecdotal, notwithstanding claims of simulation-to-real transfer.
5. The restrictions section should address the method's size and scope. How close must training and test sets be for optimum practice? How well does hardware simulation training generalize?
6. While comparisons and metrics used to evaluate MNets analyze their performance based on task/collision-avoidance success, computation time during training or testing is not mentioned.
8. Combining global and local planners can reduce poor performance owing to distribution shift. The report should explain why this is hard.
9. The two control approaches (frequency for closed-loop control) should be included in the main text.
10. The authors emphasize consistency, yet its definition is imprecise. Consistency is "how motion changes in response to a planning problem disturbance. This was not shown and needs to be clarified.
11. The model was trained on fully observable data, thus it's unclear how test-time partial observability occurs.
12. Due to consistency uncertainty, the difference between global planner and hybrid planner is unclear, making the document have some vague representations.





**Best Paper Nomination:**

No

---

> ### Author Response · Authors · 2022-08-26
> **Response to Meta Review (1/2)**
>
> We would like to thank the meta reviewer for the careful and thorough summary of our paper, its strengths, and its weaknesses. We have corrected our paper to incorporate more detail on each of these points--as well as the points addressed by the reviewers--but provide succinct summaries below. We would like to correct one statement in the meta review in which you state that our performance is comparable to MPNets without a collision checker. As shown in Table 2, our model outperforms MPNets in terms of success rate by 46% when trained and tested within the wider ranging, complex environments we have generated. While we are aware that this work has limitations (which we attempt to address below), we believe the significant improvement over prior work and strong results in complex settings demonstrates its benefit to the community at large.
>
> 1. We use the same randomized procedural generation to generate planning problems (environment, start, goal) in both train and test. While they are drawn from the same distribution, the test set’s environments, starts, and goals are all unique from those in the training data. To make the distribution transparent, we will detail the precise parameter ranges to the appendix, as well as open source the data generation code upon publication. We will also include a more informative figure demonstrating the distribution of environments in the main body of the paper ([https://bit.ly/3c7GGQD](https://bit.ly/3c7GGQD)).
> 2. MPiNets is an instantaneous policy that assumes a static world at the time of inference. If the scene changes between inference steps, the policy will react accordingly. If the environment is continually changing--as is often the case in dynamic settings--MPiNets implicitly approximates the dynamic movement as a sequence of static motions. When the scene changes are slow, this assumption works well. When the changes are fast, it does not. We ran an experiment to demonstrate with 1000 dynamic tabletop problems at four different speeds--stationary, slow, medium, and fast. Each problem was chosen to be solvable when stationary. For the static, slow, medium, and fast speeds, the success numbers are 100%, 88.1%, 57.4%, and 28.3%, respectively. Please see the discussion in the main body of our updated draft for more detail.
> 3. We have corrected these statements in the paper. We now talk about RRT's value for determining feasible paths, omit our erroneous claim about search-based planners' relative speed--instead focusing on their discrete nature-- and have corrected the reviewer's comments on our Global Planning pipeline.
> 4.  Our goal with our hardware trials was to illustrate that this technique can transfer to hardware, but in a physical system there are many interacting components. In our evaluations, our goal was to isolate our network architecture to fairly evaluate it. We do not make claims that this is ready for immediate deployment in a production environment. We have amended the language to make it clear that we provide qualitative demonstrations, rather than rigorous trials.
> 5. We have added this discussion to the limitations, explaining that we expect performance to degrade as inference environments deviate from the training distribution. We have added two experiments to demonstrate this. In one, we replace all tabletop objects in our test set with similar-sized meshes from the YCB dataset. On these trials, the success rate drops from 94.67% to 88.33%. We have also tested adding $0$-meaned Gaussian noise to the point clouds with varying standard deviations ($\sigma$). The network's performance is roughly constant with noise increasing up to $\sigma=3cm$ , when success drops to 89.28%. You can see the decaying performance with increased noise in this graph https://bit.ly/3dMFgva.
> 6. MPNets took 15 hours to train on our dataset of 3.27M demonstrations. We included inference time for all methods in our tables of success rates. We have also included our hardware specifications for inference to the Appendix.

---

> > ### Author Response · Authors · 2022-08-26
> > **Response to Meta Review (2/2)**
> >
> > We wish to reply to the Meta Review thoroughly and were not able to fit our responses into a single box. Here are the rest of our responses.
> >
> > 7. We trained a model using a combination of both experts and observed averaging behavior in the policy. That is, on the test set of problems solvable by the Global Planner, the combined policy has target convergence rate between the networks trained with demonstrations from each method individually--97.17% (combined) vs 87.72% (hybrid) vs. 97.94% (global)-- and collision rate between them as well--18.56% (combined) vs 11% (hybrid) vs 21.94% (global). Compared with the model trained on Hybrid Expert data (our primary model), we attribute the improved target performance to the more complete target coverage in the data set of Global Planner demonstrations and the increased collisions due to the increased noise in the Global Planner data.
> > 8. We have moved this description into the main paper.
> > 9. We agree that this definition is vague, but have included a more thorough discussion in the Appendix, but it remains a qualitative assessment used to provide intuition for our empirical results. This discussion further defines consistency as 1) expert quality and 2) repeatability of the planner. In terms of quality, prior work in imitation learning [https://arxiv.org/pdf/2108.03298.pdf] has demonstrated minimal path length to be a component of expert quality, and our Hybrid Planner produces shorter end effector paths.  In terms of repeatability, the hybrid planner's sampling component is in a lower-dimensional space, which typically would lead to faster convergence to an optimal path. Additionally, the smoothing component of the Hybrid Planner is deterministic whereas the smoother used for the Global Planner is not.
> > 10. This is a well-studied outcome of our model structure. Our model employs PointNet++ [Qi et. al. Neurips, 2017] as the point cloud encoder. PointNet++ has several PointNet [Qi et. al. CVPR 2016] layers. The original PointNet paper describes how it can be trained with fully observed point clouds and used for inference in partially observable settings with minimal performance drop. Our environments only ever have partial occlusion. If essential scene geometry was fully occluded, we would not expect MPiNets to perform well.
> > 11. This is related to our response to #9, but we feel that learning in task space is beneficial for several reasons. First, we think that it can produce motion more intuitive to human partners or operators. Second, goals in manipulation are typically represented in end effector task space [https://arxiv.org/pdf/2106.01352, https://arxiv.org/pdf/2103.14127]. In a closed loop setting with a moving target, the traditional process of using IK to map task to configuration space can produce highly variable configurations, especially around obstacles.